# PRISM - A Multi-Dimensional Verification Approach to Mitigate Hallucinations in Chain-of-Thought Reasoning

## Abstract

In recent years, Chain-of-Thought (CoT) verification has emerged as a critical research direction. However, existing approaches largely focus on the quality of intermediate reasoning or final answer correctness, while hallucinations arising from the initial stage of question understanding remain underexplored. To address this gap, we propose a unified framework—PRISM (**P**rogressive **R**easoning with **I**nstructional and **S**trategic **M**ulti-dimensional Verification) that jointly tackles all three aspects. We introduce a Commonsense-Augmented Progressive Instructional Reasoning (CPIR) method, designed to alleviate condition hallucination while utilizing commonsense to capture relevance between conditions and questions. Then we develop Multi-Dimensional Heterogeneous Collaborative Verification (MHCV), which strategically validates reasoning chains from multiple perspectives to enhance intermediate reasoning quality and question comprehension, thereby mitigating different types of hallucinations. In addition, we propose a Discard-Weighted Voting mechanism to overcome the limitations of traditional voting methods in multi-dimensional verification. Experimental results demonstrate that PRISM consistently improves verification accuracy across conditions, logical reasoning, and question comprehension, yielding more reliable reasoning chains and higher final-answer accuracy compared to strong CoT baselines.

## 1 Introduction

Step-by-step reasoning is a fundamental capability in both analytical thinking and problem-solving. As early as ancient Greece, philosophers like Socrates emphasized guarding against implicit fallacies through rigorous questioning step by step. In recent years, CoT prompting has advanced the reasoning capability of LLMs to a new level Wei et al. (2022); Cheng et al. (2024); Zhang et al. (2024), significantly enhancing the effectiveness of reasoning chains in addressing complex problems across a wide range of domains Li et al. (2025); Mitra et al. (2024); Hu et al. (2025); Nong et al. (2024); Liao et al. (2025); Sanwal (2025).

Although CoT significantly improves LLM reasoning, its reliability remains inconsistent. As observed by Arcuschin et al. Arcuschin et al. (2025), once an error occurs, subsequent steps may propagate flawed premises, leading to hallucinations Chu et al. (2024); Zhao et al. (2023); Feng et al. (2023). This has led to growing interest in verifying CoT-generated reasoning. Verification methods are broadly divided into forward and backward approaches. Forward verification follows the original reasoning flow, validating steps incrementally or holistically Chu et al. (2024); Weng et al. (2022); Zhu et al. (2024). However, as CoT chains grow longer, it tends to reproduce the original reasoning path Zhang & Xiong (2025), making it hard to detect question misinterpretation. Backward verification starts from the final answer to infer original premises Jiang et al. (2024), mimicking human reverse reasoning. Though more robust for complex tasks, it often overlooks intermediate reasoning steps and tends to reflect only a partial understanding of the question Weng et al. (2022); Ranaldi & Pucci (2025); Deb et al. (2024). Therefore, existing methods struggle to balance the question comprehension, intermediate reasoning quality and final answers' accuracy.

To address these challenges, we propose **Commonsense-Augmented Progressive Instruction Reasoning (CPIR)** in the reasoning stage, which mitigates condition hallucination by extracting key

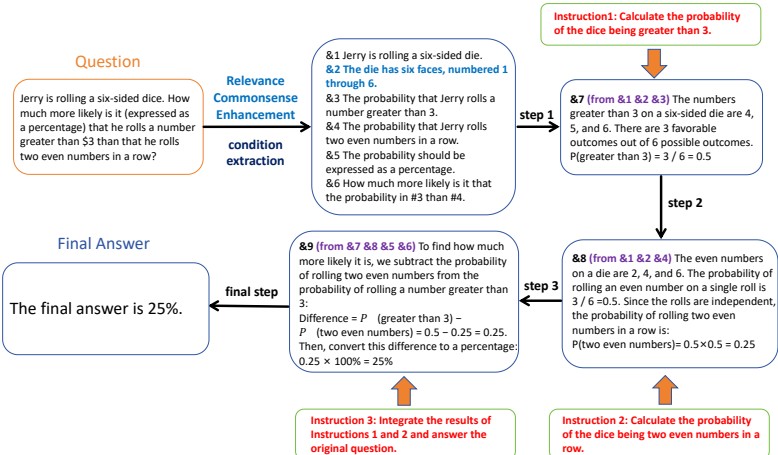

Figure 1: An example of Commonsense-Augmented Progressive Instruction Reasoning

conditions and generating progressive instructional lists, while leveraging commonsense to capture and make explicit the relational links between conditions and the target question. For verification, inspired by ancient Greek step-by-step questioning, we introduce the **Multi-Dimensional Heterogeneous Collaborative Verification (MHCV)** method. Forward verification detects conditional and logical inconsistencies, while backward verification infers the original question from progressive instructions to identify question misunderstandings. Unlike previous studies, we address hallucination issues from both forward and backward directions, with particular emphasis on question comprehension hallucination—where the LLM misunderstands or forgets the true meaning of the original question during reasoning—a challenge that has received limited attention in prior work.

We present a unified pipeline that jointly operationalizes forward and backward verification with a dedicated module for question-comprehension hallucination. Compared with deductive reasoning approaches Ling et al. (2024); Zhu et al. (2024), backward verification that reconstruct key conditions in questions (e.g., Self-Verification Weng et al. (2022)), and tool-based or retrieval-based approaches (e.g., Verify-and-Edit Zhao et al. (2023)), our framework differs in:

1. We demonstrate that LLMs can inherently generate the commonsense needed to capture the relations linking the given conditions to the target question. By making such relations explicit and introducing progressive instructions, our method mitigates implicit errors and enables structured reasoning to support effective verification.

2. We propose a unified multi-dimensional hallucination verification framework that integrates both forward and backward verification. This approach effectively addresses condition hallucination, logical errors, and question misinterpretations, balancing question understanding, intermediate reasoning quality, and final-answer accuracy.

3. We introduce a *Weighted Discard Voting* strategy that selectively retains partially correct reasoning instead of discarding entire chains for minor issues, thereby enhancing the reliability of final answers.

## 2 RELATED WORK

**Chain-of-Thought Prompting**    CoT prompting has significantly improved the reasoning ability and interpretability of LLMs in question answering tasks Wei et al. (2022); Chu et al. (2024); Li et al. (2025); Cheng et al. (2024); Sanwal (2025). Several studies Zhou et al. (2022); Lyu et al. (2023); Jung et al. (2022); Qian et al. (2022) proposed a question translator that decomposes a complex query into simpler sub-questions. Similarly, the condition translator Zhu et al. (2024) breaks down conditional logic for easier downstream reasoning. However, these decomposition strategies

often lack interpretability and offer limited guarantees on the correctness of the reformulated components Huang et al. (2025); Kumar et al. (2025). To address these issues, our method incorporates commonsense-augmented conditional decomposition and progressive instruction generation.

**Verification of Chain-of-Thought**    CoT verification can be broadly classified into forward and backward methods. Forward approaches rely on either external tools or LLMs. Tool-based methods Inaba et al. (2023); Wu et al. (2025); Jiang et al. (2025) use code interpreters, calculators, or search engines, while retrieval-based ones Zhang et al. (2023); Ni et al. (2025); Chen et al. (2025); Zhang et al. (2025) check factual validity. However, these methods face issues in robustness and security. LLM-based strategies rely on the model's own reasoning capabilities, using self-iteration, self-feedback, or stepwise validation Madaan et al. (2024); Paul et al. (2023); Ling et al. (2024). Despite the great progress, forward verification still struggles to detect implicit fallacies and often fails to capture misunderstandings arising from question comprehension. Backward verification simulates human reverse-checking by masking key information and checking for consistency in inferred reasoning Jiang et al. (2024); Weng et al. (2022). While this provides a complementary perspective, it is computationally expensive and often overlooks holistic reasoning quality. Furthermore, because it conditions on the final answer, backward verification often reflects only a partial understanding of the question—reconstructing premises that are compatible with the answer but not faithful to the original wording or constraints, and thus overlooking implicit requirements and step-level inconsistencies Ranaldi & Pucci (2025); Deb et al. (2024). To overcome these limitations, we propose a multi-dimensional verification framework to more comprehensively address various hallucinations.

**Voting Strategic**    Majority voting is widely used to select the final answer but may not generalize well across different tasks. Some approaches Wang et al. (2025); Zelikman et al. (2022); Miao et al. (2023) score reasoning paths via auxiliary models to improve quality, at the cost of increased computational load. Others Zhu et al. (2024); Vacareanu et al. (2024) apply multi-dimensional filtering before voting, but over-filtering can discard valid chains with minor flaws, reducing reliability. To address this, we propose a Discard-Weighted Voting strategy, which uses verification feedback to assign discard scores and is more suitable for multi-dimensional verification screening.

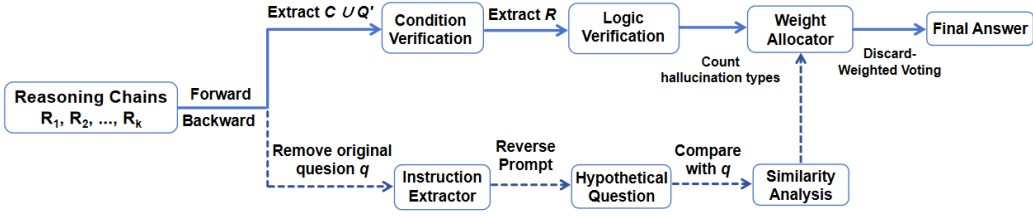

Figure 2: Architecture of Multi-Dimensional Heterogeneous Collaborative Verification

## 3    THE PROPOSED METHOD

In this section, we present the PRISM framework for reasoning and verification in Chain-of-Thought. In the reasoning stage, a Condition Display Module leverages commonsense to make conditions and condition–question relations explicit, while an Instruction Generator produces progressive instruction lists tailored to the given question. In the verification stage, we introduce MHCV, a multi-dimensional framework that verifies hallucinations across three complementary aspects. Finally, Discard-Weighted Voting is proposed to select the most reliable candidate answer.

### 3.1    COMMONSENSE-AUGMENTED PROGRESSIVE INSTRUCTION REASONING

Prior methods decompose complex problems into sub-information through a translator Zhu et al. (2024); Wu et al. (2024); Ling et al. (2024); Zhou et al. (2022), thereby simplifying the reasoning process. As illustrated in Figure 1, the *Condition Display Module* extracts the necessary conditions and integrates only the commonsense knowledge that links the conditions to the question, while discarding any irrelevant ones. Then, the *Instruction Generator* produces progressive instructions for intermediate steps, ensuring that each step moves in the correct direction.

We define a complete CoT reasoning process as a tuple $(Q, C, R, A)$, where:

- $Q$ denotes the question, consisting of the final query statement and its associated conditions.
- $C$ denotes the commonsense knowledge linking conditions to the question;
- $R$ denotes the intermediate reasoning process;
- $A$ denotes the final answer derived from the reasoning.

Inspired by Deductive Verification of Chain-of-Thought Reasoning Ling et al. (2024), we eliminate redundant information to minimize potential interference with the reasoning. Specifically, the question $Q$ is decomposed into:

$$Q' = \{b_1, b_2, \ldots, b_m, q\}$$

where $\{b_1, b_2, \ldots, b_m\}$ are the minimal background conditions required to answer the question, and $q$ is the final query statement.

Subsequently, the final inquiry $q$ is enriched with relevant commonsense knowledge, resulting in a set. Our method leverages LLM's own capabilities to supplement commonsense without relying on external interfaces, thereby improving both efficiency and security (see Appendix A.5 for details):

$$C = \{c_1, c_2, \ldots, c_p \mid q\}$$

In cases where no supplementary commonsense is deemed necessary, the model outputs "No supplementary commonsense is needed", and $C$ is treated as an empty set. The LLM then performs reasoning based on the combined input $C \cup Q'$.

During the reasoning process, an *Instruction Generator* assigns a progressive instruction to each intermediate reasoning step. This yields a sequence of intermediate reasoning steps:

$$R = \{(i_1, g_1, r_1), (i_2, g_2, r_2), \ldots, (i_n, g_n, r_n)\}$$

where $i_n$ is the $n$-th progressive instruction, $g_n$ is the index set of conditions required for the $n$-th step, and $r_n$ represents the content of the $n$-th intermediate reasoning step.

### 3.2 MULTI-DIMENSIONAL HETEROGENEOUS COLLABORATIVE VERIFICATION FRAMEWORK

Unlike traditional methods that rely solely on forward or backward verification, the MHCV framework combines both to leverage their strengths, as shown in Figure 2. Forward verification detects condition hallucination and logical errors in intermediate steps, while backward verification identifies issues in question comprehension. Beyond improving final answer accuracy, MHCV focuses on enhancing the overall quality of the reasoning chain by ensuring logical coherence and reinforcing the model's comprehension of the question.

#### 3.2.1 FORWARD CONDITIONAL AND LOGIC VERIFICATION METHOD

Forward verification follows the direction of CoT reasoning to examine various types of errors and hallucinations in intermediate reasoning steps. This method consists of two stages:

In the first stage, given the set of intermediate reasoning steps $R = \{(i_1, g_1, r_1), (i_2, g_2, r_2), \ldots, (i_n, g_n, r_n)\}$, the progressive instructions $i_k$ $(1 \leq k \leq n)$ are removed, yielding a simplified reasoning set $R' = \{(g_1, r_1), (g_2, r_2), \ldots, (g_n, r_n)\}$. Subsequently, the decomposed question set $Q'$ is used to extract the foundational background conditions $Q' - q = \{b_1, b_2, \ldots, b_m\}$, which, combined with $C$, form the complete condition set $(Q' - q) \cup C$. Each intermediate reasoning step $(g_k, r_k)$ $(1 \leq k \leq n)$ is verified against this condition set, which is progressively augmented with any newly inferred conditions. The verification checks whether:

- the referenced condition indices are correct,
- the use of those conditions is logically grounded,
- the supplemented commonsense aligns with real-world physical or social principles.

The verification outcome for the $k$-th intermediate reasoning step is as follows:

$$\mathcal{V}(g_k, r_k \mid (Q' - q) \cup C) = \begin{cases} \text{true,} & \text{if conditions are met} \\ \text{false,} & \text{otherwise} \end{cases}$$

In the second stage, logical reasoning verification assesses whether each intermediate reasoning step maintains logical coherence. This verification consists of two types: for mathematical reasoning tasks, it detects arithmetic or logical errors within candidate chains; for linguistic reasoning tasks, it identifies semantic inconsistencies. Both types of verification are performed via few-shot prompting. Let $L_k(r_k)$ $(1 \leq k \leq n)$ denote the verification result of the $k$-th reasoning step, which represents the forward verification outcome for step $k$.

$$F_k = \mathcal{V}(g_k, r_k \mid (Q' - q) \cup C) \wedge L_k(r_k)$$

The entire candidate chain passes forward verification if and only if all intermediate reasoning steps individually pass their respective forward verification.

Unlike prior works Weng et al. (2022); Zhao et al. (2023); Ling et al. (2024); Kumar et al. (2025) that decompose problems through external translators or treat reasoning holistically, our approach introduces three key innovations: (i) implicit information is systematically made explicit through a Condition Display Module, (ii) commonsense is directly supplemented by the LLM to capture condition–question relations without relying on external resources, and (iii) progressive instructions are generated to guide reasoning in a controllable, step-by-step manner.

### 3.2.2 Reverse Instructional Question Backtracking Verification Method

Although verification can ensure conditions, commonsense, and logical coherence, it cannot prevent errors from an initial misinterpretation of the question, arising from semantic ambiguity or gradual drift in reasoning. Such hallucinations are often missed by current verification methods.

To address this issue, we propose **Reverse Instructional Question Backtracking Verification**. Unlike backward methods that merely compare final answers, our approach abstracts away conditions and reasoning details, focusing instead on the instructional trajectory. By extracting intermediate instructions and using them to reconstruct the predicted question, we evaluate semantic alignment with the original question, thereby mitigating question-comprehension hallucination[1] in CoT reasoning.

To verify question consistency, we reconstruct the original question from the instructional flow extracted from sampled reasoning chains. The verification process is detailed in **Algorithm 1**.

The purpose of Reverse Instructional Question Backtracking Verification is to mitigate question-comprehension hallucination. Unlike approaches that evaluate only final answer correctness, our method examines whether the reasoning trajectory aligns with the original intent. By abstracting away numerical values and detailed steps, it enhances verification efficiency and reduces deviations from the intended question.

---

**Algorithm 1** Reverse Instructional Problem Backtracking Verification

---

**Input:** Original question $q$, decomposed question $Q' = \{b_1, b_2, \ldots, b_m, q\}$, commonsense knowledge $C$
**Output:** Verification result (Pass / Fail)
  1: Let $B \leftarrow Q' \setminus \{q\}, \quad I \leftarrow \emptyset$
  2: **for** $t = 1$ to $k$ **do**
  3:     Generate a candidate reasoning chain $R_t = \{(i, g, r)\}$ using prompt $(B \cup C)$
  4:     Extract instruction list $I_t \leftarrow \{i \mid (i, g, r) \in R_t\}$
  5:     $I \leftarrow I \cup I_t$
  6: **end for**
  7: Generate predicted question $q' \leftarrow$ LLM.generate$(I, B \cup C)$
  8: score $\leftarrow$ LLM.evaluate_semantic_equivalence$(q', q) \in [0, 1]$
  9: Let $\tau$ be a similarity threshold tuned on validation data (set 0.8 in our experiment)
 10: **if** score$(q', q) > \tau$ **then**                    ▷ For the setting of $\tau$ and detailed scoring, see Appendix A.7
 11:     **return** Pass
 12: **else**
 13:     **return** Fail
 14: **end if**

---

[1]For a detailed explanation and examples of question-comprehension hallucination, see Appendix A.4.

### 3.2.3 DISCARD-WEIGHTED VOTING

Most CoT reasoning methods adopt *Majority Voting* to select the most frequent candidate answer, while others Wang et al. (2025); Zhu et al. (2024) employ auxiliary models or multi-dimensional filtering to refine the outputs. However, in multi-dimensional verification, these strategies face two key limitations: (1) stricter screening often filters out more candidate chains, reducing the overall sample size and weakening the reliability of voting; (2) chains with minor deviations—such as mis-referenced conditions or slight phrasal variations—can still produce correct answers, yet Majority Voting may discard them entirely.

To address these issues, we propose **Discard-Weighted Voting** strategy, which assigns different weights to various types of hallucinations in order to compute a discard score for each reasoning chain. A higher score indicates more severe hallucinations and lower reliability. Given $k$ candidate chains, we count the frequency of each answer to form a frequency list $answer\_list = \{(a_1, \text{num}_1), (a_2, \text{num}_2), \ldots, (a_m, \text{num}_m)\}$. Let $a_j$ denote the $j$-th candidate answer, and $n_j$ represent the frequency with which answer $a_j$ appears among the $k$ candidate chains. $m$ is the total number of distinct answers. Based on the frequency counts, we sort answers in descending order and retain the top two with the highest vote counts, as we observe that correct answers overwhelmingly concentrate in the top two candidate answers when sampling 10 reasoning chains:

$$new\_answer\_list = \{(a^{(1)}, N_1), (a^{(2)}, N_2)\}$$

Let $N_1$ and $N_2$ denote the frequencies of the top two candidate answers $a^{(1)}$ and $a^{(2)}$ respectively. In exceptional cases in which several candidate answers tie for the second-highest frequency, we include all such answers in the candidate list.

For all reasoning chains associated with the top candidate answer $a^{(1)}$ and $a^{(2)}$, both forward and backward verifications are performed. Each reasoning chain is evaluated for the presence of the following three types of hallucinations: (1) condition hallucination – including condition index errors, incorrect condition references, omissions, additions, and supplementary commonsense knowledge that is inconsistent with real-world facts. This category is assigned a weight $\omega_1$, initially set to 0.4 in our experiments[2]. (2) Logical reasoning hallucinations – for mathematical reasoning tasks, this includes arithmetic logic errors; for linguistic reasoning tasks, it includes semantic inconsistencies. This category is assigned a weight $\omega_2$, also set to 0.4. (3) question-comprehension hallucination – referring to cases where the reasoning chain misinterprets or fails to retain key information from the original question. This category is assigned a weight $\omega_3$, set to 0.2.

The normalized discard score of each candidate answer is computed as follows:

$$Score_a = \frac{1}{N_a} \sum_{c \in C_a} \sum_{i=1}^{3} \omega_i \cdot \varphi_i(C)$$

$N_a = |C_a|$ denotes the number of reasoning chains associated with candidate answer $a$. $\varphi_t(C)$ is the indicator function for the $t$-th type of hallucination in the $i$-th reasoning chain: $\varphi_t(C) = 1$ if any intermediate reasoning step in the chain exhibits a type-$t$ hallucination, and $\varphi_t(C) = 0$ otherwise. Finally, the candidate answer with the lower weighted discard score is selected as the final verified answer:

$$\hat{a} = \arg \min_{i \in \{1,2\}} Score_{a^{(i)}}$$

## 4 EXPERIMENTS

In this section, we evaluate the effectiveness of CPIR method and MHCV framework across diverse tasks. We first assess individual and overall verification performance on five datasets. Then we compare final answer accuracy against strong baselines. Finally, ablation studies are conducted

---

[2]The verification experiments for the weight allocation of $\omega_1$, $\omega_2$, $\omega_3$ are detailed in Appendix A.2.

Table 1: Comparison of verification accuracy between MHCV and CoT method across five datasets for condition, logic, question-comprehension and overall hallucinations. Verified Type denotes the type of verification: condition, logic, and backward refer to accuracy in detecting condition hallucination, logical errors, and question-comprehension hallucination respectively; overall indicates the accuracy of verifying the full reasoning chain.

| Verification Method | Verified Type | GSM8K | AddSub | AQuA-RAT | Last Letter | Date | Average |
|---|---|---|---|---|---|---|---|
| CoT (two-shot) | Overall | 50% | 58% | 52% | 50% | 50% | 52.0% |
| MHCV (one-shot) | Condition | 64% | 74% | 72% | 82% | 72% | 72.8% |
| | Logic | 84% | 90% | 68% | 76% | 62% | 76.0% |
| | Backward | 60% | 64% | 76% | 80% | 66% | 65.6% |
| | **Overall** | **68%** | **66%** | **74%** | **66%** | **60%** | **66.8%** |

Table 2: Final Answer Accuracy comparison among CPIR w/o MHCV verification and three strong baselines. Each method uses GPT-3.5-Turbo as the backbone and generates 10 candidate reasoning chains per question.

| Methods | Arithmetic Tasks | | | Semantic Tasks | |
|---|---|---|---|---|---|
| | GSM8K | AddSub | AQuA-RAT | Last Letter | Date |
| Self-Consistency Wang et al. (2022) | 75.74% | 82.28% | 59.40% | 80.44% | 57.99% |
| Faithful CoT Lyu et al. (2023) | 75.80% | 88.35% | 61.80% | —[3] | **73.50%** |
| Self-Harmonized CoT Jin & Lu (2024) | 78.50% | 87.10% | 55.50% | 81.30% | —[3] |
| Deductive Verification Ling et al. (2024) | 86.01% | 93.54% | 69.49% | 92.60% | 71.45% |
| **CPIR (w/o Verification)** | **87.57%** | **93.92%** | 69.29% | **94.67%** | 72.48% |
| **CPIR + MHCV** | 86.81% | 93.16% | **71.26%** | 94.00% | 71.93% |

to demonstrate the effectiveness of each module. For experiments on voting weights and k value selection, see Appendix A.2 and A.6.

## 4.1 DATASET

To evaluate both arithmetic and semantic reasoning capabilities, we test our approach on several benchmark datasets. For arithmetic reasoning, we use AddSub Hosseini et al. (2014), a dataset of simple addition and subtraction problems, GSM8K Cobbe et al. (2021), which contains 8,500 high-quality middle school math word problems, and AQuA-RAT Ling et al. (2017), a more challenging multiple-choice dataset covering algebra, proportions, and sequences. For semantic reasoning, we include Last Letter Concatenation, which requires extracting and concatenating the last letter of each word in a sentence to assess symbolic manipulation, and Date Understanding (DU) Srivastava et al. (2022), which evaluates the model's ability to comprehend and reason about temporal information in date-related queries.

## 4.2 EXPERIMENTAL PREPARATION

**Verification test of MHCV framework** To evaluate the effectiveness of forward and backward verification methods in MHCV framework, we manually annotated 50 reasoning chains per dataset, totaling 250 question–answer pairs. Each candidate answer underwent condition verification, logic verification, reverse instructional backtracking, and overall CoT accuracy assessment, yielding 1000 verification results. For each dataset, the 50 samples consisted of 25 correct answers and 25 hallucinated answers. The hallucination cases were manually annotated according to their natural occurrences and distributions in real datasets, ensuring comprehensive coverage of all observed types. Specifically, the annotated set covers four categories: (1) condition hallucination, (2) logical error, (3) question-comprehension hallucination, and (4) mixed-type hallucinations, with diverse instances included in each category to enhance representativeness and robustness of the evaluation.

**Model selection** GPT-3.5-Turbo serves as the primary backbone in our experiments, where both CPIR and MHCV are implemented through CoT prompting without task-specific annotations or fine-tuning. To assess the generalizability of our methods, we additionally employ DeepSeek V3-0324 as an alternative backbone (Detailed results are presented in Appendix A.1).

Table 3: Final answer accuracy on five datasets in the ablation study. *w/o Commonsense Enhancement* removes the commonsense-augmented module. *w/o Backtracking Verification* removes the backward verification component, with discard weights for condition hallucination and logic reasoning set to 1:1. *w/o Discard-Weighted Voting* uses Majority Voting instead.

| Method | GSM8K | AddSub | AQuA-RAT | Last Letter | Date |
|---|---|---|---|---|---|
| **PRISM (ours)** | **86.81%** | **93.16%** | **71.26%** | **94.00%** | **71.93%** |
| *w/o Commonsense Enhancement* | 84.34% | 90.12% | 66.54% | 93.33% | 69.38% |
| *w/o Backtracking Verification* | 86.05% | 92.66% | 65.35% | 93.33% | 70.30% |
| *w/o Discard-Weighted Voting* | 85.44% | 92.15% | 68.11% | **94.00%** | 70.30% |

**Model hyperparameters**    We set the maximum output length of LLMs to max_tokens_len = 2048 and the temperature to 0.7. During the reasoning phase, the number of candidate reasoning chains is fixed at $k = 10$. For the selection of k, we balance the inference accuracy and computational cost. The impact of k on reasoning accuracy can be found in Appendix A.6. In the Discard-Weighted Voting stage, three types of hallucination weights are assigned: $\omega_1 = 0.4$ for condition hallucination, $\omega_2 = 0.4$ for logical reasoning inconsistency, and $\omega_3 = 0.2$ for question-comprehension hallucination.

## 4.3 Verification Ability of MHCV Method

In this experiment, we manually annotate 50 reasoning chains per dataset following the CPIR format to assess the effectiveness of MHCV framework for each hallucination type, with 25 fully correct and 25 containing one or more errors: condition hallucination, arithmetic/semantic logic hallucination, or question-comprehension hallucination. We then evaluate whether the MHCV method can accurately identify the error type in each annotated case. We compare its overall verification performance with a standard two-shot CoT baseline to assess whether the proposed multi-dimensional framework improves verification ability.

In the evaluation of MHCV method, verification is performed across three dimensions of hallucinations. A reasoning chain is considered correct only if it successfully passes all three verification. If any single verification fails, the entire chain is deemed incorrect. In this section, we compare the MHCV framework with a two-shot CoT baseline, in which the model is guided through step-by-step verification using dedicated prompts for condition, commonsense and logical reasoning verification.

As shown in Table 1, the verification accuracy improves significantly across all datasets when using the MHCV framework. This improvement can be attributed to MHCV's ability to mitigate diverse types of hallucinations in CoT reasoning. Notably, the backward verification demonstrates strong effectiveness in addressing question-comprehension hallucination, enabling the model to better retain and interpret the original intent of the question—particularly in longer or more complex reasoning scenarios.

In contrast, the performance on the AddSub dataset is relatively modest. This is likely due to the dataset's inherent simplicity—both in conditional structure and computational complexity—where the advantages of condition decomposition, instruction backtracking, and multi-dimensional verification contribute less noticeably to overall reasoning quality.

## 4.4 Accuracy Test of CPIR and MHCV

In Section 4.3, we demonstrated that MHCV significantly enhances the quality and accuracy of CoT reasoning. In this section, we further evaluate the final answer accuracy of reasoning chains generated using the CPIR method and subsequently verified through the MHCV framework. To assess final answer correctness, we adopt two evaluation settings: (1) **CPIR-only**: This setting uses the CPIR method without any verification. The final answer is selected using Majority Voting, with the number of candidate reasoning chains set to $k = 10$; (2) **CPIR + MHCV**: This setting

---

[3]The original work of Faithful CoT Lyu et al. (2023) did not report results on the *Last Letter* task, while Self-Harmonized CoT Jin & Lu (2024) did not report results on the *Date* task. Neither work specified the prompt format for these datasets, so the corresponding entries are left blank.

Table 4: Token consumption and backward verification accuracy across three backward reasoning methods on the 250-sample annotated dataset. *Full Backward* reconstructs the entire original question using the full reasoning chain. *Forward–Backward Reasoning* masks a key condition and infers it backward from the final answer. *Our Method* employs progressive instruction-based backward reconstruction.

| Method | Prompt Tokens ↓ | Output Tokens ↓ | Total Tokens ↓ | Accuracy ↑ |
|---|---|---|---|---|
| Full Backward | 199,051 | 28,919 | 227,970 | 66.40% |
| Forward–Backward Reasoning Jiang et al. (2024) | 123,051 | 46,110 | 169,161 | 73.60% |
| **Our Method** | **110,801** | **23,942** | **134,743** | **79.20%** |

integrates the CPIR method with the MHCV verification framework, replacing Majority Voting with the proposed Discard-Weighted Voting strategy.

We compare the performance of these two configurations against several representative methods that have demonstrated strong empirical results and were widely adopted in recent years. These include: Self-Consistency Wang et al. (2022), Faithful CoT Lyu et al. (2023), Self-Harmonized CoT Jin & Lu (2024), and Deductive Verification Ling et al. (2024), all of which emphasize generating or validating reliable and faithful reasoning paths. The comparative results are presented in Table 2.

After applying MHCV, we observe a slight decline in final-answer accuracy, consistent with prior work Ling et al. (2024). This decline mainly comes from minor reasoning flaws that typically do not change the final answer but are flagged by stricter verification. To address this, our Discard-Weighted Voting mechanism reduces undue penalties on such benign errors.

Interestingly, on AQuA-RAT, MHCV improves final-answer accuracy. This can be attributed to the dataset's higher rate of question-comprehension hallucination, where reverse verification better aligns reasoning with the intended question. Moreover, the dataset's complexity induces longer reasoning chains with more intermediate steps, increasing pruning during verification. In such cases, Majority Voting tends to discard more valid chains, reducing answer reliability—a limitation effectively addressed by our proposed voting strategy.

## 4.5 COMPUTATIONAL COST OF BACKWARD VERIFICATION

**Backward verification cost analysis** Since the proposed MHCV framework integrates both forward and backward verification to detect condition hallucinations, logical hallucinations, and question-comprehension hallucinations, its computational cost is inherently higher than that of conventional verification schemes. In particular, backward verification requires reconstructing the original conditions, question semantics, and an additional set of reasoning instructions, resulting in a substantial increase in token usage. Therefore, this section compares the token cost of our instruction-based backward reconstruction method with two baseline backward-verification approaches.

We include two backward verification baselines:

(1) *Forward-backward verification Jiang et al. (2024)*, which masks a single condition and infers it from the final answer to determine answer correctness;

(2) *Full backward reconstruction*, which utilizes the entire reasoning chain to infer the semantic content of the original question and generate a full-step instructional trace.

In the experiment, we use the manually annotated set of 250 candidate reasoning chains, focusing exclusively on evaluating backward question-understanding verification. Unlike previous datasets, this set does not include condition or logical hallucination cases. Consequently, no interference occurs between different hallucination types, and the backward verification accuracy of MHCV is expected to be higher than in earlier evaluations. Figure 1 and Table 4 report the total token consumption and backward verification accuracy of the three methods on the annotated dataset.

**Results and analysis** The results show that reconstructing the original question from the full reasoning trace significantly increases the number of prompt tokens, leading to a total token cost approximately 69.19% higher than our method. Moreover, providing excessive reasoning-chain information does not improve verification accuracy; instead, it reduces accuracy, suggesting that

effective backward reconstruction depends on step-wise instructional cues rather than full-chain reasoning content.

For the forward–backward reasoning method, its verification target differs from ours: it infers masked conditions rather than reconstructing question semantics. Although it requires even more tokens than our approach, its accuracy is noticeably lower. These findings demonstrate that our instruction-based backward reconstruction method achieves a more favorable balance between efficiency and accuracy, outperforming both baselines in token cost and verification reliability.

### 4.6 ABLATION EXPERIMENT

We conduct ablation studies to evaluate the contribution of each MHCV component. As shown in Table 3, we examine three variants: (1) w/o Commonsense Enhancement; (2) w/o Backward Verification; and (3) w/o Discard-Weighted Voting. In the last setting, if all candidate chains are rejected or multiple answers tie, the model outputs the answer with the highest vote count.

We observe that the AQuA-RAT dataset exhibits the most substantial performance drop, likely due to its complex reasoning structures and higher incidence of question-comprehension hallucination. Moreover, applying Majority Voting within multi-dimensional verification often leads to a limited pool of valid voting samples, thereby reducing the reliability of the final answer selection.

## 5 LIMITATIONS

While our method effectively mitigates multi-dimensional hallucinations, two limitations remain:

(1) As shown in Table 1, verification of question-comprehension hallucinations lags behind other checks. This gap is attributable to the volatility of semantic scores generated by LLMs in backward verification. Future work could further explore the setting of varying $\tau$ and conduct more experiments using a broader range of LLMs. For a detailed analysis, see appendix A.10.

(2) We currently apply a unified framework across various tasks, but task-specific adaptations may yield better performance. For example, AQuA-RAT is more prone to question-comprehension hallucination, while others rely more on commonsense or instruction-level cues. Thus, optimal strategies and voting weights may vary by task and warrant further exploration.

(3) Our current experiments are conducted on relatively standard reasoning benchmarks, and do not yet cover more challenging settings such as MATH datasets with substantially deeper and more compositional reasoning. In addition, the manually annotated hallucination set used for analysis is still limited in size. In future work, we plan to evaluate PRISM on more complex benchmarks, expand the manually annotated dataset, and perform a more fine-grained statistical analysis of different hallucination categories so that the annotation can better match realistic task distributions.

## 6 CONCLUSIONS

This study aims to improve question comprehension, intermediate reasoning quality and final answer accuracy in Chain-of-Thought through a multi-dimensional perspective. To achieve this, we propose a two-stage framework - **PRISM**. The first stage leverages LLM's own capabilities to enhance commonsense and generate progressive instructions to support subsequent verification. The second stage introduces the Multi-Dimensional Heterogeneous Collaborative Verification (MHCV) framework to detect condition, logic and question comprehension hallucinations. Additionally, we design the Discard-Weighted Voting strategy to improve final answer accuracy under multi-dimensional filtering. Experiments show that our framework improves question comprehensive ability while enhancing inference quality and final answers' accuracy. Our approach is dedicated to providing a multi-stage and multi-dimensional framework for future chain-of-thought studies.

## ETHICS STATEMENT

This work adheres to the ICLR Code of Ethics. All experiments are conducted on publicly available benchmark datasets (GSM8K, AddSub, AQuA-RAT, Date Understanding, and Last Letter), which do not involve human subjects, private information, or sensitive attributes. For model inference, we employed the OpenAI GPT-3.5-Turbo API and the DeepSeek V3-0324 API as backbones. These APIs were accessed in compliance with their respective terms of service, and no personally identifiable or sensitive user data were involved. The proposed framework operates solely on model-generated outputs, without introducing external personal or proprietary information. Therefore, this study does not raise direct ethical concerns regarding privacy, security, or fairness. We acknowledge the potential societal impacts of advancing large language model reasoning and have carefully documented our methodology to support transparency and integrity in research.

## REPRODUCIBILITY STATEMENT

We have taken multiple steps to ensure the reproducibility of our results. Detailed dataset descriptions and preprocessing steps are provided in Appendix A.3. Model hyperparameters, experimental configurations, and candidate selection settings are described in Section 4 and Appendix A.6. Verification procedures and evaluation metrics are explained in Sections 3 and 4, with further implementation details given in Appendices A.2 and A.4. All datasets used are publicly available, and representative code snippets for answer extraction are included in Appendix A.3. Anonymous source code and experiment scripts are provided in the supplementary materials, with detailed usage instructions documented in the accompanying README file. Together, these materials allow other researchers to reproduce our experiments and verify our findings.

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

# A  APPENDIX

## A.1  PERFORMANCE OF PRISM ON OPEN SOURCE LLM

In addition to evaluating the inference and verification capabilities of the proprietary model GPT-3.5-turbo on five benchmark datasets, we further examine the generalizability of PRISM by incorporating a recent popular open-source language model, Deepseek V3-0324. During both the reasoning and verification stages, responses are generated using Deepseek V3-0324, adhering to the Progressive Instructional Reasoning format illustrated in Figure 1.

Table 5: Comparative performance of Deepseek V3-0324 and GPT-3.5-turbo on five reasoning datasets under PRISM framework.

| Model Name | GSM8K | AddSub | AQuA-RAT | Last Letter | Date Understanding |
|---|---|---|---|---|---|
| Deepseek V3-0324 | 96.21% | 96.96% | 88.98% | 100.00% | 91.33% |
| GPT-3.5-turbo | 86.81% | 93.16% | 71.26% | 94.00% | 71.93% |

As shown in Table 5, Deepseek V3-0324 demonstrates notable improvements in accuracy across all five datasets, attributed to its strong arithmetic and semantic reasoning capabilities. The performance gains are especially pronounced on more challenging tasks, further validating the generalizability of PRISM in both the reasoning and verification stages under the chain-of-thought paradigm.

## A.2  FINAL ANSWER'S ACCURACY OF DIFFERENT DISCARD WEIGHTS

As shown in Table 6, the first three rows present configurations where each type of hallucination is individually assigned a dominant weight. These settings yield lower final answer accuracy on most datasets compared to the 2:2:1 ratio, suggesting that the three verification types are complementary and that overemphasis on any single type weakens overall validation. The fourth row evaluates an equal-weight setting (1:1:1), which also underperforms, indicating that the three types contribute unequally to final answer quality and highlighting the benefit of strategically weighted verification.

An exception is observed in the Last Letter dataset, where certain configurations yield higher final answer accuracy. This is likely due to its shallow reasoning structure, which involves surface-level text extraction without complex logic or implicit commonsense. As a result, condition and logic hallucination are rare, while question-comprehension hallucination—such as mistaking the task for extracting first or middle letters—are more common. In this context, assigning greater weight to backward verification helps correct such misinterpretations and improves final answer accuracy.

Table 6: Impact of different weight configurations in Discard-Weighted Voting on final answer accuracy across five datasets. dc:dl:db = 2:2:1 corresponds to weights of 0.4 (condition), 0.4 (logic), and 0.2 (backward). The remaining configurations are scaled proportionally. The last row (in bold) represents the configuration adopted in our main experiments.

| Weight Distribution | GSM8K | AddSub | AQuA-RAT | Last Letter | Date |
|---|---|---|---|---|---|
| dc:dl:db = 8:1:1 | 86.43% | 92.41% | 68.90% | 94.67% | 70.03% |
| dc:dl:db = 1:8:1 | 85.90% | 92.66% | 70.08% | 94.67% | 70.84% |
| dc:dl:db = 1:1:8 | 86.73% | 92.91% | 67.32% | **95.33%** | 70.57% |
| dc:dl:db = 1:1:1 | 86.35% | 92.15% | 68.50% | **95.33%** | 69.75% |
| dc:dl:db = 1:2:2 | 86.58% | 92.15% | 69.29% | **95.33%** | 70.57% |
| dc:dl:db = 2:1:2 | 86.28% | 91.90% | 68.50% | 94.67% | 69.48% |
| **dc:dl:db = 2:2:1** | **86.81%** | **93.16%** | **71.26%** | 94.00% | **71.93%** |

## A.3 DATASET DETAILS AND FINAL ANSWER EXTRACTION

To comprehensively evaluate the reasoning and verification capabilities of PRISM across a range of tasks, we conduct experiments on five benchmark datasets. Specifically, GSM8K, AddSub, and AQuA-RAT are employed to assess arithmetic reasoning, while Date Understanding and Last Letter are used to evaluate semantic logical reasoning. A detailed overview of each dataset, including sample size, the format of answers, and type of task, is provided in Table 7.

Table 7: Overview of the five benchmark test datasets used for evaluation, including sample size, answer format, and type of task.

| Dataset | Sample Size | Answer Format | Type of Task |
|---|---|---|---|
| GSM8K | 1319 | Number | Arithmetic reasoning |
| AddSub | 395 | Number | Arithmetic reasoning |
| AQuA-RAT | 254 | A/B/C/D/E | Arithmetic reasoning |
| Date Understanding | 369 | M/D/Y (date) | Semantic logical reasoning |
| Last Letter | 150 | String | Semantic logical reasoning |

Due to the varying answer formats across datasets, we standardized the model outputs to ensure consistent and accurate final answer extraction. To this end, we instructed the model to append a summary sentence containing the final answer, following a designated delimiter ####. This approach facilitates reliable parsing of the final answer. A representative code snippet for the extraction function is shown in Table 8.

Table 8: Key codes of extraction functions for different reasoning tasks

| Extracted Content | Key Code of Extraction Function |
|---|---|
| number | |
| | ```match = re.search(r"####\s*([\d,]+(?:\.\d+)?)", ans_str)``` |
| A/B/C/D/E | |
| | ```match = re.search(r"####\s*([A-E])\b", ans_str)``` |
| date information | |
| | ```match = re.search(r"####\s*(.+)", ans_str)``` |
| letter | |
| | ```match = re.search(r'the_answer_is\s+"([^"]+)', ans_str, re.IGNORECASE)``` |

## A.4 BACKWARD VERIFICATION OF QUESTION-COMPREHENSION HALLUCINATION

Question-comprehension hallucination refers to a model's failure to accurately comprehend the true intent of a question. This represents a deeper and more nuanced form of hallucination, often challenging to detect through conventional chain-of-thought (CoT) verification methods. There are two primary scenarios in which this type of hallucination tends to occur:

- **Semantic Misinterpretation:** The model initially fails to capture the correct semantic meaning of the question, resulting in a misinterpretation of what is actually being questioned.

- **Contextual Drift:** As the reasoning process unfolds, the model gradually loses focus on the original question or forgets its content altogether, causing its reasoning trajectory to deviate from the intended objective.

In comparison to condition hallucination and logic hallucination, question-comprehension halluci-nation are more deeply embedded and significantly harder to detect—even with a dedicated verifi-cation module. This form of error often stems from the model's inability to maintain a coherent and anchored understanding of the question throughout the reasoning process.

A representative example of question-comprehension hallucination is presented in Table 9.

The Instruction Generator extracts the instruction associated with each intermediate reasoning step and compiles them into a sequence. During backward verification, the original question—"How many years will it take before he starts earning money on the lemon tree?"—is masked. The model is then prompted to infer the most plausible original question based solely on the extracted instructions.

As shown in Table 8, the model hypothesizes: "How many years will it take for Carlos to break even on the lemon tree?" A semantic comparison reveals a mismatch in intent. This reasoning chain is flagged as hallucinated due to the model's misinterpretation: while Carlos breaks even in year 12, when total profit equals the initial investment, he only begins to earn net profit in year 13, when cumulative profit exceeds the cost. The model's conflation of "start earning" with "break even" leads to an incorrect answer.

## A.5 COMMONSENSE SUPPLEMENT IN CONDITION DISPLAY MODULE

The focus of the Condition Display Module is to integrate and refine the conditional information provided by the original problem, collecting as much problem-relevant content as possible while fil-tering out irrelevant parts, so that the reasoning chain can rely on a more complete and accurate set of conditions. A key step in this process is the supplementation of commonsense—not as generic back-ground facts, but as relational knowledge that connects decomposed conditions to each other and to the target question. By explicitly surfacing these hidden links, the module reduces the likelihood of implicit fallacies during reasoning. Unlike previous commonsense augmentation approaches, our method neither calls any external interfaces nor supplements broad background knowledge that may introduce noise. Instead, it relies solely on the large model's own capacity to generate concise com-monsense that strengthens the relational connections among conditions and between conditions and the problem statement.

First, the Condition Display Module supplements the necessary commonsense according to the re-lational structure of the problem. This involves identifying missing dependencies or associations among conditions and between conditions and the query, and then making them explicit within the structured condition list. With the advent of the latest large models such as DeepSeek R1 and Qwen 3, we believe these models are already capable of capturing most of the required relational common-sense; external APIs would not only consume more time and computation but also risk introducing irrelevant background information.

The supplementation of commonsense varies with the type of the problem. For example, when date calculation in February is involved, it can supply the relation between "leap year" and "number of days," clarifying how a background condition modifies the main query; if dice-probability calcula-tion is required, it can link the property of dice faces to the probability sub-conditions. In addition, we observe that many problems involve counting items of certain categories, where distinguish-ing relevant from irrelevant categories is crucial. For instance, when counting fruits, the original conditional information may list the number of sandwiches as a distraction; supplementing the com-monsense relation that "sandwiches are not fruits" explicitly links this condition back to the actual question, thereby preventing errors. Representative examples of such commonsense supplements are shown in Table 10.

Finally, in some cases, no additional commonsense is required because the conditions and their relations to the question are already explicit. In such cases, the large model is prompted to output "no additional commonsense needed," so as to avoid redundant information that might obscure the condition–question relationship.

### A.6 Selection of k in Candidate Chain-of-Thought

Although a larger number $k$ of candidate chains of thought makes the selection of the final answer more reliable, the choice of $k$ must also take into account computational and time costs. Multi-dimensional verification filters candidate chains from multiple aspects—conditions, commonsense, and logic—often eliminating more candidates; therefore, it is necessary to increase $k$ to ensure that a sufficient number of chains remain after screening, so that subsequent Majority Voting has enough samples for voting.

This method proposes *Discard-Weighted Voting*, which, compared with Majority Voting, does not reject an entire candidate chain because of a small error within it, thereby effectively alleviating the need to raise $k$ under multi-dimensional verification and improving the efficiency of chain-of-thought reasoning.

We sampled 50 question–answer pairs from each of the five datasets and averaged the results to evaluate the effect of different values of $k$ on the final answer accuracy; see Table 11 for details.

### A.7 Why we score question semantic similarity by LLMs

In our *Reverse Instructional Problem Backtracking Verification*, we use a large language model (LLM) to score the semantic equivalence between the predicted question $q'$ and the original question $q$, rather than relying on cosine similarity over sentence embeddings (e.g., SBERT). We choose LLM-based scoring because it is more robust to rephrasings, word order changes, and syntactic variations, and thus better at recognizing "same meaning, different phrasing."

As an illustration, original $q$: "How many U.S. states border Colorado?"; predicted $q'$: "How many states are adjacent to Colorado in the United State?" The two differ in wording ("border" vs. "adjacent") and structure, but target the same meaning (the count of neighboring states). For such cases, LLM-based equivalence scores are typically higher and more reliable than static-embedding cosine similarity, which can be sensitive to lexical and positional shifts.

As also reflected in Table 1, backward verification for question-comprehension hallucinations trails other checks. This gap aligns with the volatility of LLM-produced semantic scores discussed above, which in turn makes the choice of a universal threshold $\tau$ uncertain. In this paper we report $\tau = 0.8$ based on validation data for our main setup, but we do not claim universality; as noted in our limitations, future work should examine varying $\tau$ and broaden evaluation across different LLMs.

### A.8 Analysis of Accuracy Degradation After Applying the MHCV Framework

As shown in Table 2, the final-answer accuracy decreases slightly on most datasets after applying the MHCV framework. This observation is expected: MHCV is designed not to maximize final accuracy, but to prioritize the rejection of reasoning chains that exhibit hallucinations in any intermediate step. By enforcing multi-dimensional verification on conditions, logical derivations, and question comprehension, the framework favors *reliable* chains over those that merely arrive at the correct answer by coincidence or through flawed reasoning.

Several boundary cases illustrate this effect. Some reasoning chains reference condition indices incorrectly while maintaining correct content; others contain minor logical inconsistencies but, through compensatory errors, still reach the correct final answer; still others introduce irrelevant or semantically redundant reasoning steps. Although such chains may yield the correct solution, they are structurally unreliable. MHCV therefore tends to classify them as untrustworthy, intentionally filtering out chains that are correct but not credible. This selective rejection mechanism explains the small drop in final-answer accuracy observed after applying MHCV.

To mitigate the impact of minor hallucinations on the final decision, we introduced the Discard-Weighted Voting mechanism. Ablation experiments confirm its effectiveness in balancing reliability and accuracy. In the current version, hallucination severity is approximated using the count of hallucination types detected. For future work, we plan to improve Discard-Weighted Voting by refining hallucination taxonomy and assigning differentiated weights to each hallucination type. This would allow the model to distinguish between "minor" issues—such as misnumbered condition references

or harmless redundant statements—and "severe" issues such as factually incorrect commonsense or arithmetic reasoning errors. Such finer-grained weighting is expected to strengthen the reliability of MHCV while reducing its negative impact on final-answer accuracy.

## A.9 MANUALLY ANNOTATED VERIFICATION DATASET

We construct two manually annotated datasets, each containing 250 questions paired with model-generated reasoning chains. The two datasets share exactly the same set of questions but differ in the hallucination types they evaluate. The first dataset includes reasoning chains that may exhibit any subset of the three hallucination categories—condition hallucination, logic hallucination, and question-comprehension hallucination (ranging from 0 to 3 types). The second dataset, in contrast, contains only question-comprehension hallucinations. Its purpose is to evaluate the computational cost and accuracy of different backward-verification methods.

**Dataset for multi-dimensional hallucination analysis.** To systematically evaluate MHCV's ability to identify all three types of hallucinations, we construct a 250-sample dataset with fine-grained human annotations. Specifically, we randomly sample 50 questions each from GSM8K, AddSub, AQuA-RAT, Last Letter, and Date Understanding. For every question, we generate ten candidate reasoning chains using DeepSeek V3, from which annotators select 50 fully correct reasoning chains and their corresponding 50 questions. We then manually annotate the ground-truth hallucination labels and ensure per-dataset balance by including 25 chains with entirely correct intermediate reasoning and 25 chains containing at least one hallucination type.

Each reasoning chain is annotated along four dimensions:

- **Condition Hallucination:** Whether referenced values, entities, condition indices, or condition contents appear in the original problem or correspond to valid commonsense; whether supplemented commonsense aligns with natural or physical laws. Any fabricated or incorrect reference is labeled as `false`.
- **Logic Hallucination:** Whether intermediate steps follow valid arithmetic or semantic logic. Violations such as incorrect numerical operations or ill-defined deductive jumps are labeled as `false`.
- **Question-Comprehension Hallucination:** With the final answer masked, whether the reasoning chain remains aligned with the original question. Deviations such as answering a different question or omitting critical aspects result in a `false` label.
- **Overall Correctness:** A holistic judgment integrating all three dimensions as well as the correctness of the final answer.

**Dataset for backward-verification analysis.** The second dataset uses the same 250 questions but is annotated solely for backward question-comprehension hallucination. For each question, half of the reasoning chains are labeled as entirely correct, while the other half contain explicit comprehension hallucinations. This dataset is used to evaluate the computational overhead and accuracy of our backward-verification method compared with baseline approaches.

**Annotation protocol.** All samples were independently annotated by five researchers with backgrounds in mathematics and NLP. Prior to annotation, annotators reviewed typical hallucination patterns prevalent in each dataset to ensure broad coverage and adherence to realistic error distributions. The final 250 annotated samples cover a diverse range of errors including numerical miscalculations, missing or fabricated conditions, faulty derivation structures, and multiple forms of question misinterpretation. This unified benchmark supports all experiments in the paper concerning MHCV verification accuracy and the computational cost of backward verification.

## A.10 ANALYSIS OF SEMANTIC-SCORING INSTABILITY IN BACKWARD VERIFICATION

In the backward-verification module, we abstract the reasoning chain into a sequence of structured instructions and compute a semantic similarity score $s(q, \hat{q})$ between the reconstructed question $\hat{q}$

and the original question $q$. This score is used to determine whether a question-comprehension hallucination has occurred. Unlike condition and logic verification—which rely on discrete, explicitly checkable constraints—semantic equivalence is an open-ended natural-language matching problem. It is inherently sensitive to variations in phrasing, paraphrasing, word order, redundancy, and stylistic differences. As a result, the distribution of semantic scores can vary considerably across datasets, tasks, and even samples within the same dataset. Semantically equivalent paraphrases may receive unexpectedly low scores, while partially mismatched reformulations may be assigned artificially high scores. Such instability can cause fixed-threshold decisions based on $\tau$ to behave inconsistently, reducing the robustness of backward verification.

Moreover, semantic similarity operates in a continuous "soft matching" space and lacks the discrete, structurally observable constraints that make condition and logic verification highly reliable. Consequently, both the false-positive and false-negative rates of semantic scoring are substantially higher than those of the other two detectors. In practice, this manifests in two characteristic failure modes: (i) *over-penalization*, where reasoning chains that correctly solve the original problem are erroneously rejected due to benign paraphrasing differences; and (ii) *under-penalization*, where chains that partially deviate from the intended question are mistakenly accepted. This instability further propagates into the final Discard-Weighted Voting process, motivating our choice to assign a relatively small weight to the backward-verification dimension. Our sensitivity analysis across different values of $\tau$ (reported in the appendix) highlights this dependence on the score distribution.

To address these issues, we plan to improve the semantic-scoring mechanism in future work. One direction is to adopt an adaptive thresholding strategy, where $\tau$ is selected using a small development set tailored to the target task and dataset distribution. Another direction is to incorporate more structured signals—such as condition coverage, target-quantity alignment, and reasoning-path consistency—to complement the inherently noisy semantic score. These enhancements aim to reduce distributional instability while preserving the interpretability benefits of backward verification.

### A.11 Weight Allocation for Condition, Logic, and Question-Comprehension Verification

In Appendix A.2, we experimented with different weight combinations for condition hallucination, logic hallucination, and question-comprehension hallucination, and evaluated their impact on the final-answer accuracy across all datasets. Empirically, the configuration $\omega_{\text{cond}} : \omega_{\text{logic}} : \omega_{\text{backward}} = 2 : 2 : 1$ achieved the strongest performance on the arithmetic and semantic–logical reasoning tasks considered in this paper. In this section, we provide a theoretical explanation for why this ratio performs best, and discuss how the weight assignment may change for future tasks.

To analyze the relative importance of the three hallucination detectors, we treat each detector $V_t$ ($t \in \{1, 2, 3\}$, corresponding respectively to condition verification, logic verification, and question-comprehension verification) as a binary classifier for the latent correctness label $y \in \{0, 1\}$, where $y = 1$ denotes a reliable reasoning chain and $y = 0$ denotes an unreliable one. For detector $V_t$, we denote its binary decision function as $\phi_t(C) \in \{0, 1\}$, where $C$ is the observed reasoning chain and $\phi_t(C) = 1$ means that the detector judges the chain as *passing* (i.e., reliable), while $\phi_t(C) = 0$ means the chain is judged as *failing*.

We define the false-positive rate (FPR) of detector $V_t$ as

$$\alpha_t = \Pr\big(\phi_t(C) = 1 \mid y = 0\big),$$

which measures how often the detector incorrectly marks an *unreliable* reasoning chain ($y = 0$) as passing.

Similarly, the false-negative rate (FNR) is defined as

$$\beta_t = \Pr\big(\phi_t(C) = 0 \mid y = 1\big),$$

which measures how often the detector incorrectly rejects a *reliable* reasoning chain ($y = 1$).

We compute an overall error level as

$$\varepsilon_t = \tfrac{1}{2}(\alpha_t + \beta_t),$$

and define an "observability" indicator

$$\delta_t = 1 - \alpha_t - \beta_t,$$

which quantifies how well detector $V_t$ separates reliable from unreliable reasoning chains. A higher $\delta_t$ indicates stronger discriminative ability and lower misclassification noise.

Condition verification and logic verification rely primarily on discrete, symbol-level constraints such as value referencing, entity matching, and arithmetic consistency. These checks are highly observable and tend to have low $\alpha_t$ and $\beta_t$, leading to larger $\delta_t$ and smaller $\varepsilon_t$. In contrast, question-comprehension verification must reconstruct the underlying question from a reasoning trace and evaluate semantic equivalence. This process is inherently noisier and more sensitive to paraphrasing, ambiguity, and information redundancy, resulting in significantly higher $\varepsilon_{\text{backward}}$ and lower $\delta_{\text{backward}}$. Refer to appendix A.10 for cause analysis. Overall, the detectors typically satisfy

$$\delta_{\text{cond}} \approx \delta_{\text{logic}} > \delta_{\text{backward}}, \qquad \varepsilon_{\text{cond}} \approx \varepsilon_{\text{logic}} < \varepsilon_{\text{backward}}.$$

Under the standard assumption that detectors are approximately conditionally independent given the true label, the posterior log-odds

$$\log \frac{\Pr(y = 0 \mid \phi_1, \phi_2, \phi_3)}{\Pr(y = 1 \mid \phi_1, \phi_2, \phi_3)}$$

can be expressed as a linear function of the detector outputs:

$$\log \frac{\Pr(y = 0 \mid \phi_1, \phi_2, \phi_3)}{\Pr(y = 1 \mid \phi_1, \phi_2, \phi_3)} \approx \text{const} + \sum_{t=1}^{3} g_t\big(\phi_t(C)\big),$$

where each increment $g_t(\cdot)$ is monotonically related to the detector's error level $\varepsilon_t$. Detectors with lower $\varepsilon_t$ contribute more strongly to the posterior decision boundary, while detectors with higher $\varepsilon_t$ contribute less. The weighted discard score used in MHCV,

$$\text{Score}_a(i) = \sum_{t=1}^{3} \omega_t \cdot \phi_t(C_i),$$

can thus be viewed as a parametric linear approximation to the above log-odds. Because condition and logic verification exhibit significantly lower noise compared to question-comprehension verification, a natural ordering emerges:

$$\omega_{\text{cond}} \approx \omega_{\text{logic}} > \omega_{\text{backward}}.$$

The empirical choice of $2 : 2 : 1$ is a coarse but effective instantiation of this ordering. It reflects the fact that condition and logic verification provide similarly reliable evidence and should dominate the decision process, while question-comprehension verification—despite being essential for capturing semantic-level deviations—contributes noisier signals and therefore should be assigned a smaller weight. Our ablation results in Appendix A.2 confirm that this asymmetric allocation improves overall stability and leads to consistently higher final-answer accuracy.

For future tasks, however, the optimal weight distribution may not remain fixed at $2 : 2 : 1$. In domains such as open-domain QA or multi-turn dialogue, question-comprehension errors may be more prevalent, and the semantic verification component can often be improved through task-specific training or adaptation, thereby reducing $\varepsilon_{\text{backward}}$ and increasing $\delta_{\text{backward}}$. In such settings, weights may naturally shift toward more balanced configurations, such as $1 : 1 : 1$ or even $1 : 1 : 2$, depending on the relative reliability of each detector. From this perspective, the $2 : 2 : 1$ configuration used in this paper is best interpreted as a task-aligned instantiation rather than a universal hyperparameter.

## A.12 USAGE OF LARGE LANGUAGE MODELS

We used LLM solely to aid in polishing the writing of this paper after the main content had been completed. The LLM was not used for ideation, retrieval of related work, experimental design, analysis, or any other aspect of the research process. All conceptual, technical, and experimental contributions are original and were carried out independently.

Table 9: Illustrative Example of Question-Comprehensive Hallucination

| | |
|---|---|
| **Question** | Carlos is planting a lemon tree. The tree will cost $90 to plant. Each year it will grow 7 lemons, which he can sell for $1.5 each. It costs $3 a year to water and feed the tree. How many years will it take before he starts earning money on the lemon tree? |
| **Ground Truth** | He makes $10.5 selling lemons each year because $7 \times 1.5 = 10.5$. He earns $7.5 each year from the lemon tree because $10.5 - 3 = 7.5$. It will take 12 years to pay off the tree because $90/7.5 = 12$. He will make money in year 13 because $12 + 1 = 13$. #### 13 |
| **Model Answer** | #1. The cost to plant the lemon tree is $90. #2. The lemon tree grows 7 lemons each year, which can be sold for $1.5 each. #3. It costs $3 a year to water and feed the tree. #4. How many years will it take before Carlos starts earning money on the lemon tree? #5. Carlos will start earning money when revenue exceeds total cost. #6. (by #2) Step 1: Calculate the annual revenue from selling lemons. (Instruction 1) Number of lemons per year = 7 Price per lemon = $1.5 Annual revenue = $7 \times 1.5 = 10.5$ #7. (by #3) Step 2: Calculate the annual recurring cost of watering and feeding. (Instruction 2) Annual maintenance cost = $3 #8. (by #1, #6, #7, #5) Step 3: Set up the equation to find the break-even year. (Instruction 3) Let $x$ be the number of years. Total revenue over $x$ years = $10.5x$ Total cost = Initial planting cost ($90) + Annual maintenance cost ($3x$) Break-even point: $10.5x = $90 + $3x$ $10.5x$ - $3x = $90 $7.5x = $90 $x = $90 / $7.5 = 12$ #9. (by #4, #8) So the answer to this question is after 12 years Carlos will begin to earn profit. (Hallucination) #### **12** **(The correct answer is 13)** |

Table 10: Examples of different types of example questions and their corresponding commonsense supplements.

| Common-sense Type | Example Question | Commonsense Supplement |
|---|---|---|
| Time & Calendar | In 2023, a streaming pass covers every day of February plus the first 3 days of March. In 2024, the same pass covers every day of February plus the last 4 days of January. A family buys one pass each year and uses it on every covered day. Across both years, how many total days are covered by the passes? | February has 28 days in common years and 29 days in leap years. |
| Item Category | A deli packs meal kits: each kit has 4 apples, 3 bananas, 2 oranges, 5 cupcakes, and 1 bottle of juice. Nora buys 3 kits, then during the week she eats 7 pieces of fruit, gives 5 pieces of fruit to a neighbor, and throws away 2 cupcakes that got squished. Counting only fruit items as single pieces and ignoring non-fruit, how many pieces of fruit does Nora have left? | Apples, bananas, and oranges are fruit; cupcakes and bottled juice are not counted as fruit pieces. |
| Units & Measurements | A craft shop sells beads in mixed containers: 2 boxes of one dozen beads each, 3 tubes of 25 beads, and a bulk bag holding half a gross. During a workshop, 19 beads roll away and are lost, and a student later finds 6 of the lost beads and returns them. How many beads remain after the workshop? | One dozen equals 12; a gross equals 144 (so half a gross is 72). |
| Currency & Value | Kaylee has 2 rolls of quarters (40 quarters per roll), 1 roll of nickels (40 nickels per roll), and loose coins: 7 quarters, 6 dimes, 4 nickels, and 12 pennies. She pays $23.87 for supplies using only these coins, handing over the largest values first until she reaches or exceeds the price, and receives no change. How many cents does she have left afterward? | Quarter = 25¢, dime = 10¢, nickel = 5¢, penny = 1¢. |

| Common-sense Type | Example Question | Commonsense Supplement |
|---|---|---|
| Days & Scheduling | A weekend workshop runs for 6 consecutive weeks and meets on both Saturday and Sunday each week. The organizer cancels all meetings on the 3rd weekend, then adds make-up meetings on the next two Mondays. Later, a surprise guest gives one extra session on a Sunday during the final week. Over the whole period, how many meetings are held? | Weekend days are Saturday and Sunday (2 per weekend). |
| Mathematical Properties | A school has lockers numbered 1 through 120. Students may choose a locker only if its number is prime and not a multiple of 3 or 5 (so numbers like 3 or 5 are excluded). The student council also removes any choice above 100 to keep assignments near classrooms. How many locker numbers remain available? | A prime number has exactly two positive divisors (1 and itself), and 1 is not prime. |
| Relationships & Classification | Rina has two siblings: an older brother with 3 children (including one set of twins counted as two children) and a younger sister with 2 children. Rina's spouse's sister has 1 child, and Rina's cousin has 2 children. For a family party, Rina is bringing gifts only for her own nieces and nephews. How many gifts does she need? | Your siblings' children are your nieces/nephews; cousins' children and your in-laws' children are not your nieces/nephews. |

Table 11: Effect of different values of $k$ on the overall final answer accuracy across the five datasets

|  | $k = 1$ | $k = 3$ | $k = 5$ | $k = 7$ | $k = 10$ |
|---|---|---|---|---|---|
| **Accuracy** | 73.76% | 78.35% | 79.72% | 81.05% | 81.77% |

