# OpenReview forum: "PRISM: A Multi-Dimensional Verification Approach to Mitigate Hallucinations in Chain-of-Thought Reasoning"
_ICLR.cc/2026/Conference — ICLR 2026 Conference Withdrawn Submission_

### Official Review · Reviewer_shFo · 2025-10-21

**Soundness:** 2
**Presentation:** 2
**Contribution:** 2
**Rating:** 2
**Confidence:** 4

**Summary:**

The paper introduces a new prompting frameworking: PRISM, which aims to reduce hallucinations in long CoT reasoning. Specifically, the paper first proposes Commonsense-Augmented Progressive Instructional Reasoning (CPIR) which explicitly list commonsense knowledge, and ask related instructions in the reasoning process. The paper further introduces multi-dimensional heterogeneous collaborative verification (MHCV) which verifies the faithfulness of the reasoning process. Additionally, the paper proposes a Discard-Weighted Voting mechanism to replace majority voting.

**Strengths:**

- The problem of hallucination and faithfulness in the reasoning process, especially in long-cot scenarios, is an important problem towards more robust and stronger reasoning model; the paper provides insights on different aspects to approach the problem, including "enhancing commensense knowledge", "more fine-grained verification", etc
- The paper writing is clear and the ablation study on each of the components are presented in Table 3.

**Weaknesses:**

- I find the major weakness of the work is that it lacks a well defined problem and the proposed method is three weakly-related prompting engineering methods, each targeting a slightly different problem; It is hard to find a core contribution beside prompt engineering efforts; and lacks a deeper insights on how to solve the hallucination problem during model training.
- In Table 2, without verification results in better performance in most times, it is hard to justify why the MHCV is still necessary (although the authors mentioned that this is also observed in a previous work, it does not make sense why this is acceptable)
- The paper lacks discussion on the computational overhead given the much more complicated prompting strategy. Since the paper is an inference-only approach, it is important to compare inference time as a reference for the trade-off between performance and efficiency

**Questions:**

- If I understand correctly, in Table 1, the random performance would be 50%; does that mean that the CoT baseline achieves only random performance on most tasks? This does not seem natural to me, could you present the full prompts for all methods?

---

### Official Review · Reviewer_LSRp · 2025-10-28

**Soundness:** 3
**Presentation:** 3
**Contribution:** 3
**Rating:** 6
**Confidence:** 4

**Summary:**

This study focuses on the "problem understanding stage hallucination" that has been overlooked in existing chain-of-reasoning verification, fills the research gap of only focusing on intermediate reasoning quality or final answer correctness, and the proposed CPIR and MHCV modules form a complete logical chain covering "problem understanding - intermediate reasoning - answer verification".

**Strengths:**

1. The proposed PRISM framework is innovative in multi-dimensional verification of chain-of-reasoning hallucinations and has certain academic value overall.
2. It integrates forward verification (detecting conditional and logical errors) and reverse verification (identifying problem understanding biases), and proposes a discard-weighted voting mechanism. This mechanism avoids the excessive discarding of reasoning chains with minor flaws in traditional voting, and has promoted the methodology of chain-of-reasoning verification to a certain extent.
3. The experimental design is relatively complete. It selects two types of datasets: arithmetic reasoning (GSM8K, AddSub, etc.) and semantic reasoning (Last Letter, Date Understanding), covering different reasoning scenarios. The role of each module is verified through ablation experiments, and comparisons are made with mainstream baseline models, resulting in relatively convincing outcomes.

**Weaknesses:**

1. It only mentions "low verification accuracy of problem understanding hallucinations" and "lack of task adaptation", without in-depth analysis of the causes (such as the technical root of the volatility of LLM semantic scoring), nor does it propose specific future improvements.
2. It fails to clearly explain the annotation rules and consistency check methods when manually annotating 250 "problem-reasoning chain" samples (the objectivity of sample annotation is questionable), and the discussion depth is insufficient.

**Questions:**

None.

---

### Official Review · Reviewer_L81N · 2025-11-01

**Soundness:** 3
**Presentation:** 3
**Contribution:** 3
**Rating:** 4
**Confidence:** 3

**Summary:**

PRISM reduces hallucinations in Chain-of-Thought reasoning through a two-stage approach: first, it enhances reasoning with commonsense-augmented progressive instructions; second, it applies multi-dimensional verification combining forward checks and backward reconstruction, finalized by a weighted voting mechanism that preserves partially valid chains. Experiments on arithmetic and semantic reasoning benchmarks show that PRISM reduces diverse hallucination types and enhances verification and answer accuracy compared to strong CoT baselines.

**Strengths:**

1.While forward/backward verification exist in prior work, their integration into a unified framework to combat distinct hallucination types is a distinct contribution.
2.The paper's quality is high, evidenced by a rigorous methodological design and thorough experimentation.
3.The paper is generally clearly written. The problem statement is well-defined, and the PRISM framework is explained in a structured, step-by-step manner. The use of formal notations adds precision and architectural diagrams effectively aids comprehension.
4.Hallucination in CoT reasoning is a critical barrier to the reliable deployment of LLMs. PRISM represents a meaningful advance towards more robust and trustworthy reasoning systems.

**Weaknesses:**

1.The experimental validation on five datasets is primarily test well-defined, closed-world reasoning. The framework's effectiveness on more open-ended and complex tasks remains unproven.
2. The paper correctly identifies the volatility of LLM-based semantic scoring as a key limitation of the backward verification module. However, the solution—tuning a threshold (τ) is a brittle and non-generalizable.
3. The paper fails to quantify the totaloverhead imposed by the full PRISM pipeline. Condition, logic, backward and voting represent a substantial increase in inference time and API calls compared to standard CoT.

**Questions:**

1. If the LLM has a fundamental misunderstanding of the question, wouldn't the generated instructions also be flawed from the start, leading the entire reasoning chain astray? How does CPIR mitigate this initial misdirection?
2.When the final answer is incorrect despite passing verification, what type of error is most frequent?
3.PRISM framework involves significant computational overhead. Have you quantified the average increase in token usage or latency compared to standard CoT or a strong baseline?
4.The ablation study in A.2 suggests the optimal weights for Voting might be task-dependent. Did the dynamic weighting scheme is explored?
5.In tab.3, removing the commonsense module causes the largest accuracy drop on AQuA-RAT (71.26% → 66.54%) compared to other datasets. Why is it more dependent on commonsense supplementation than others?

---

> ### Author Response · Authors · 2025-11-20
>
> Thank you for your constructive question, and we will answer it to the best of our knowledge.
>
>  (1) If the model initially misunderstands the problem, its generated instructions may indeed deviate accordingly. However, there may be a slight misunderstanding of the roles of CIPR and MHCV. CIPR produces a concise instruction list at the outset, allowing the subsequent reasoning chain to consistently revolve around these instructions and preventing gradual drift or loss of the original intent during long reasoning. Thus, CIPR’s function is to pre-establish the reasoning path, and more importantly, to make this path explicit so that the backward verification module has a clear structural basis.
>
> (2) When an answer passes verification but is still incorrect, the errors primarily fall into two types. One involves fine-grained misunderstandings of the question that go undetected—for example, phrasings that appear semantically similar but reflect different logical goals may be misjudged as equivalent. The other involves hallucinations in conditions, commonsense, or logic that forward verification does not catch, because it follows the flawed chain’s internal reasoning trajectory. Although the paper does not provide a systematic quantitative breakdown, these two categories appear most frequently in practice.
>
> (3)This issue has already been explained in the Official Comment.
>
> (4) The ablation study shows that different tasks depend on verification signals to varying degrees. Currently, we use fixed global weights and have not introduced task-level or instance-level dynamic weighting. Since the five datasets we evaluated already perform well under fixed weights and we have not yet tested broader settings, dynamic weighting remains unexplored but is potentially valuable as future work.
>
> (5) Regarding why removing the commonsense module leads to a larger performance drop on AQuA-RAT than on other datasets, we believe this is due to the dataset’s longer problems and heavier use of distracting conditions, which makes it more dependent on commonsense-based condition linking. The commonsense module does more than supplement background knowledge—it plays a crucial role in identifying which conditions are truly relevant to the final query. Without this step, noise at the condition level becomes difficult to filter, and both reasoning and verification stages amplify the resulting errors, producing a more significant decline in accuracy.

---

### Official Review · Reviewer_KZyz · 2025-11-03

**Soundness:** 3
**Presentation:** 3
**Contribution:** 2
**Rating:** 2
**Confidence:** 3

**Summary:**

This paper proposes PRISM (Progressive Reasoning with Instructional and Strategic Multi-dimensional Verification), a framework that addresses hallucinations in Chain-of-Thought (CoT) reasoning through two main components: (1) Commonsense-Augmented Progressive Instructional Reasoning (CPIR) for the reasoning stage, and (2) Multi-Dimensional Heterogeneous Collaborative Verification (MHCV) for verification. The approach targets three types of hallucinations: condition hallucination, logical errors, and question-comprehension hallucination. The authors evaluate their method on five datasets (GSM8K, AddSub, AQuA-RAT, Last Letter, Date) using GPT-3.5-Turbo and DeepSeek V3.

**Strengths:**

* Useful Setting. It is critical to detect logic errors and question understanding.
* Clear presentation. The writing is easy to follow.
* Comprehensive experiments that include five different datasets and two backbones.

**Weaknesses:**

* Lack of simple baselines: The proposed work requires more token budgets, and a simple baseline comparison should be self-consistency. The lack of baseline methods makes me unclear how effective PRISM is.
* Limited theoretical justification for weight assignments: The choice of weights (ω₁=0.4, ω₂=0.4, ω₃=0.2) appears somewhat arbitrary. While Table 5 provides some exploration, the paper lacks a principled justification for weighting condition and logic errors equally and higher than question-comprehension errors across all tasks.
* MHCV seems less effective since Table 2 shows that MHCV most of the time reduces the performance. However, Table 3 shows that removing verification results in a performance drop. It seems the numbers in the two tables somewhat contradict each other.

**Questions:**

See above

---

### Author Response · Authors · 2025-11-20

We sincerely thank all reviewers for their thoughtful and constructive feedback. We greatly appreciate the valuable insights provided and agree with many of the points raised. We have incorporated the suggested revisions and further improved the clarity of the manuscript.

First, regarding computational overhead, since our method jointly performs forward and backward verification to detect multi-dimensional hallucination types in candidate reasoning chains, it naturally incurs more inference tokens than standard CoT. In particular, backward verification is more computationally demanding because it requires reconstructing the original question from the predicted answer. This motivates our focus on demonstrating that Reverse Instructional Backtracking is more efficient than traditional backward-verification approaches. Accordingly, in Section 4.5 we have added an experiment analyzing the computational cost and accuracy of backward verification, in which we re-annotated the previously used 250-sample human-labeled dataset. The results show that our backward-verification approach achieves clear advantages in both efficiency and verification accuracy. It is worth noting that this human-labeled dataset contains only question-comprehension hallucinations and is therefore not affected by other types of hallucinations. Consequently, the accuracy of backward verification is higher than the former experiment. Details are provided in “Dataset for backward-verification analysis” in Appendix A.9 and step 4 of the readme file in the code package.

Second, MHCV is primarily designed to mitigate hallucinations that occur within intermediate reasoning steps. The backward-verification component enhances the model’s comprehension of the original question and improves the interpretability and trustworthiness of the reasoning chain. Reviewer KZyz’s concerns about Tables 2 and 3 may stem from different assumptions regarding the experimental configuration: Table 3 removes only the backward-verification module, while forward condition verification and logic verification remain applied after CoT generation. Thus, the two tables correspond to distinct ablation settings and should not be directly compared. Regarding the slight decrease in accuracy after incorporating MHCV, we provide a more detailed analysis of the underlying factors in Appendix A.8 and propose potential directions for future improvement.

Third, we acknowledge limitations in our choice of baselines and datasets. We have added self-consistency as an additional baseline and, in Appendix A.9, we provide detailed descriptions of the 250 manually annotated samples used for verification evaluation. We also expand the analysis of verification capabilities. We concur with reviewer L81N that evaluating on more challenging datasets (e.g., MATH) would further demonstrate the effectiveness of PRISM. While we agree on the importance of such evaluations, performing full-scale comparisons across multiple baselines within the rebuttal timeframe is not feasible. However, we plan to include such evaluations in future work, and we have added this point to the Limitations section.

Fourth, in response to reviewer shFo, the proposed method is not merely a concatenation of prompts. A key contribution of our work is the systematic identification of common hallucination types in CoT—condition/commonsense hallucinations, logical hallucinations, and question-comprehension hallucinations. Our main contribution is a multi-dimensional verification framework that jointly considers the origin, progression, and outcome of the reasoning chain, thereby improving explainability. We acknowledge that our explanation of hallucination mitigation strategies can be refined, and we will revise the manuscript to provide clearer elaboration. Regarding the concern about CoT accuracy being around 50%, we note that similar findings have been reported in prior work “Detective Verification of Chain-of-Thought Reasoning”. With one-shot prompting using a correct example, models often produce the correct final answer even when intermediate steps contain hallucinations. To facilitate reproducibility, we include the exact prompt used in standard CoT verification.

For reviewer KZyz’s question about weight allocation in Discard-Weighted Voting, we analyze the underlying reasons why weights associated with condition and logic hallucination contribute more to verification accuracy than those for question-comprehension hallucination, and we provide theoretical justification and additional discussion in Appendix A.11. For reviewer L81N’s concerns regarding instability in semantic scoring, we examine cross-model generalization and outline potential improvements, as detailed in Appendix A.10.

Finally, we sincerely thank all reviewers again for their valuable suggestions. We will continue to refine the manuscript and incorporate additional experiments and analyses to further strengthen the quality and clarity of the paper.

---

### Note · Authors · 2026-01-02

I have read and agree with the venue's withdrawal policy on behalf of myself and my co-authors.